# Natural Autoantibodies in Biologic-Treated Rheumatoid Arthritis and Ankylosing Spondylitis Patients: Associations with Vascular Pathophysiology

**DOI:** 10.3390/ijms25063429

**Published:** 2024-03-18

**Authors:** Diána Simon, Dorottya Kacsándi, Anita Pusztai, Boglárka Soós, Edit Végh, György Kerekes, Monika Bodoki, Szilvia Szamosi, Gabriella Szűcs, Zoltán Prohászka, Péter Németh, Tímea Berki, Zoltán Szekanecz

**Affiliations:** 1Department of Immunology and Biotechnology, Clinical Center, University of Pécs Medical School, 7624 Pécs, Hungary; simon.diana@pte.hu (D.S.); nemeth.peter@pte.hu (P.N.); berki.timea@pte.hu (T.B.); 2Department of Rheumatology, Faculty of Medicine, University of Debrecen, 4032 Debrecen, Hungary; kacsandi.dorottya@gmail.com (D.K.); anita.pusztai01@gmail.com (A.P.); soosbogi@gmail.com (B.S.); veghe22@gmail.com (E.V.); monika.czokolyova@gmail.com (M.B.); szamosi.szilvi@gmail.com (S.S.); szucsgpafi@gmail.com (G.S.); 3Intensive Care Unit, Department of Medicine, Faculty of Medicine, University of Debrecen, 4032 Debrecen, Hungary; gkerekesg@gmail.com; 4Research Laboratory, Department of Medicine and Hematology, Semmelweis University, 1125 Budapest, Hungary; prohaszka.zoltan@med.semmelweis-univ.hu

**Keywords:** rheumatoid arthritis, ankylosing spondylitis, natural autoantibodies, anti-TNF therapy, biomarkers

## Abstract

Cardiovascular (CV) morbidity and mortality have been associated with rheumatoid arthritis (RA) and ankylosing spondylitis (AS). Natural autoantibodies (nAAb) are involved in innate immunity, as well as autoimmunity, inflammation, and atherosclerosis. There have not been any studies assessing the effects of biologics on nAAbs in RA and AS, also in relation to vascular pathophysiology. Fifty-three anti-TNF-treated RA and AS patients were included in a 12-month follow-up study. Anti-citrate synthase (CS) and anti-topoisomerase I fragment 4 (TOPO-F4) IgM and IgG levels were determined by ELISA. Ultrasonography was performed to assess brachial artery flow-mediated vasodilation (FMD), common carotid intima-media thickness (ccIMT), and arterial pulse-wave velocity (PWV). Other variables were also evaluated at baseline and 6 and 12 months after treatment initiation. Anti-TNF therapy improved FMD in RA and PWV in AS and stabilized ccIMT. TNF inhibition increased anti-CS IgM and IgG, and possibly also anti-TOPO-F4 IgG levels. Various correlation analyses revealed that nAAbs might be independently involved in autoimmunity as well as changes in inflammation and vascular pathology over time in biologic-treated patients (*p* < 0.05). We also found associations between anti-TOPO-F4 IgG and anti-Hsp60 IgG (*p* < 0.05). Baseline nAAb levels or nAAb level changes might determine changes in CRP, disease activity, FMD, PWV, and ccIMT over time (*p* < 0.05). The interplay between arthritis and inflammatory atherosclerosis, as well as the effects of anti-TNF biologics on these pathologies, might independently involve nAAbs.

## 1. Introduction

Rheumatoid arthritis (RA) and ankylosing spondylitis (AS) have been associated with increased risk for atherosclerotic cardiovascular (CV) disease (ASCVD) [1,2,3]. Early endothelial dysfunction and activation may precede these events already in the pre-clinical phase of arthritides [1,2,3].

Natural antibody (nAb)-producing B-cells have been considered an intermediate stage of evolution between innate and adaptive immunity. nAbs are immunoglobulins that are produced without antigen priming and are involved in the first line of immune defense during an infection [4,5,6]. Natural autoantibodies (nAAbs) recognize evolutionarily conserved self-structures, and these antibodies have been detected in the sera of patients with various autoimmune diseases as well as healthy individuals [4,5,6]. We have previously identified two important nAAbs recognizing a mitochondrial inner membrane enzyme, citrate synthase (CS) and topoisomerase I, a major autoantibody target in systemic sclerosis (SSc) and systemic lupus erythematosus (SLE) [4,5]. Anti-CS IgM and IgG nAAbs recognize a target nucleosome antigen in systemic autoimmune diseases [4,5]. IgM and IgG nAAbs against fragment F4 (amino acids 451-593) of topoisomerase I (TOPO-F4) have been detected in the sera of SSc and SLE patients, as well as healthy subjects [4,5]. In anti-dsDNA IgG-positive SLE patients, significantly higher levels of anti-CS and anti-TOPO-F4 IgG nAAbs were observed compared to controls. In addition, increased levels of IgM anti-CS and anti-TOPO-F4 nAAbs were detected in anti-dsDNA IgM-positive SLE patients [4]. These nAAbs have also been implicated in atherosclerosis and ASCVD [6,7,8]. For example, SSc-specific autoantibodies, such as anti-topoisomerase I, in association with microvascular pathology were found to be an independent predictive risk factor for the progression of Raynaud’s phenomenon to SSc [7]. Anti-TOPO I is also an independent risk factor for macrovascular damage in SSc [8].

Autoantibodies to heat shock proteins (Hsp), such as anti-human Hsp60 and anti-Mycobacterial Hsp65, have also been characterized as nAAbs [6,9,10,11,12,13,14,15,16,17,18]. Anti-Hsp autoantibodies have also been implicated in inflammation, autoimmunity, and atherosclerosis [6,9,10,11,12,13,14,15,16,17,18]. Antibodies to Hsps trigger early atherosclerosis in various animal models [10,11]. Moreover, anti-Hsp60 antibodies have been correlated with ASCVD in humans [6,14,15]. Among inflammatory rheumatic diseases, nAAbs to Mycobacterial Hsp65 and human Hsp60 have been detected in RA, AS, SLE, and SSc [6,9,12,14,15,16,17,18]. Moreover, anti-Hsp autoantibodies have been implicated in early atherosclerosis associated with various autoimmune-rheumatic diseases [14,15].

Thus, nAAbs, including anti-CS, anti-TOPO-F4, and anti-Hsp, might form a link between innate immune responses against infectious agents, adaptive immunity associated with autoimmune diseases, and inflammatory atherosclerosis [4,5,14,15].

We have not found any studies on the effects of targeted therapies, including biologics, on the production of anti-CS or anti-TOPO-F4 nAAbs in RA, AS, or any other autoimmune-inflammatory rheumatic diseases. In this very same cohort, we assessed the possible effects of one-year anti-tumor necrosis factor α (TNF-α) therapy on anti-Hsp60 IgG levels; however, biological therapy did not change anti-Hsp60 levels over time [9].

In order to further elucidate the effects of TNF-α inhibition on nAAb production, we assessed anti-CS and anti-TOPO-F4 nAAb levels over time during one-year anti-TNF therapy. As both RA and AS have been associated with CV pathologies and these nAAbs might also influence the CV system, we correlated our results with various biomarkers of vascular pathophysiology as well. These biomarkers included brachial artery flow-mediated vasodilation (FMD), common carotid intima-media thickness (ccIMT), and aortic pulse-wave velocity (PWV), indicators of endothelial dysfunction, overt atherosclerosis, and arterial stiffness, respectively [1,19]. We also assessed vascular laboratory biomarkers in association with nAAbs [9,19]. Moreover, we used our previous anti-Hsp60 data [9] to assess possible correlations between anti-CS/anti-TOPO-F4 and anti-Hsp60 levels in this cohort. This study might improve our understanding of the role of nAAbs in arthritides as well as arthritis-associated inflammatory atherosclerosis.

## 2. Results

### 2.1. Effects of TNF Inhibition on Disease Activity and Systemic Inflammation

The effects of ETN/CZP treatment in the mixed cohort of RA + AS, as well as in RA and AS patients, on disease activity (DAS28-CRP) and CRP levels, have been published earlier [9,19]. Therefore, as we cannot present the original data again, we just briefly mention that anti-TNF therapy significantly decreased DAS28-CRP, BASDAI, and CRP in these patient subsets after 6 and 12 months of treatment (*p* < 0.05).

### 2.2. Effects of TNF Inhibition on Vascular Pathophysiology

The effects of TNF-α blockade on FMD, ccIMT, and PWV in the full RA + AS cohort were published earlier [19], therefore we cannot present the original data again. In brief, a significant, transient improvement of brachial artery FMD was observed after 6 months (*p* < 0.005), which was not sustained until 12 months of therapy. On the other hand, ccIMT did not change but remained stable over time. Finally, PWV significantly decreased after 12 months (*p* < 0.05) [19].

On the other hand, we have not yet published the FMD, ccIMT, and PWV data in RA and AS separately. In RA, FMD transiently improved after 6 months (10.4 ± 6.4%) compared to baseline (7.0 ± 5.2%; *p* = 0.012). However, FMD again increased after 12 months (9.5 ± 5.4%; *p* = 0.204 vs. baseline). Moreover, ccIMT did not change after 6 months (0.59 ± 0.08 mm; *p* = 0.502) and 12 months (0.56 ± 0.10 mm; *p* = 0.550) versus baseline (0.56 ± 0.08 mm). Similarly, PWV remained stable after 6 months (8.24 ± 1.58 m/s; *p* = 0.455) and 12 months (7.81 ± 3.19 m/s; *p* = 0.191) compared to baseline (8.32 ± 2.29 m/s) (Figure 1).

In AS, FMD showed a tendency for a non-significant increase after 6 months (9.1 ± 4.1%; *p* = 0.196) and 12 months (8.9 ± 5.0%; *p* = 0.408) compared to baseline (7.7 ± 3.5%). Similarly, ccIMT remained stable after 6 months (0.51 ± 0.07 mm; *p* = 0.148) and 12 months (0.53 ± 0.09 mm; *p* = 0.070) versus baseline (0.49 ± 0.07 mm). In contrast, PWV did not change after 6 months (6.09 ± 0.87 m/s; *p* = 0.215), but significantly decreased after 12 months (5.76 ± 1.23 m/s; *p* = 0.013) compared to baseline (6.52 ± 1.26 m/s) (Figure 1).

### 2.3. Effects of Anti-TNF Therapy on Circulating nAAb Levels

In the mixed cohort of RA and AS patients, anti-CS IgM concentrations significantly increased after both 6 months (481.6 ± 349.6 units; *p* = 0.045) and 12 months (518.7 ± 335.4 units; *p* < 0.001) compared to baseline (406.4 ± 250,5 units). Similarly, anti-CS IgG levels also increased after 6 months (145.9 ± 78.6 units; *p* = 0.006) and 12 months (143.4 ± 86.2 units; *p* = 0.023) compared to baseline (126.5 ± 73.2 units). Anti-TOPO-F4 IgM levels did not change significantly after 6 months (269.6 ± 151.2 units; *p* = 0.966) and 12 months (316.4 ± 191.7 units; *p* = 0.206) compared to baseline (267.6 ± 149.9 units). On the other hand, while anti-TOPO-F4 IgG autoantibody levels did not change after 6 months (27.5 ± 32.5 units; *p* = 0.204), they increased after 12 months (32.3 ± 35.9 units; *p* = 0.030) in comparison to baseline (21.3 ± 29.0 units) (Figure 2).

In the RA patient subset, anti-CS IgM levels did not change after 6 months (495.1 ± 381.7 units; *p* = 0.184); however, they significantly increased by 12 months (545.8 ± 335.1 units; *p* = 0.002) versus baseline (421.7 ± 244.9 units). Anti-CS IgG levels showed a non-significant increase after 6 months (139.8 ± 84.6 units; *p* = 0.058) and 12 months (141.9 ± 99.0 units; *p* = 0.108) compared to baseline (127.2 ± 83.3 units). Anti-TOPO-F4 IgM levels did not change after 6 months (262.7 ± 157.7 units; *p* = 0.879) and 12 months (323.5 ± 210.1 units; *p* = 0.204) in comparison to baseline (255.5 ± 128.4 units). Anti-TOPO-F4 IgG concentrations also did not change after 6 months (21.0 ± 27.2 units; *p* = 0.925) but significantly increased after 12 months (28.6 ± 25.6 units; *p* = 0.042) in comparison to baseline (19.0 ± 26.0 units) (Figure 2).

Finally, in AS patients, serum anti-CS IgM concentrations did not change after 6 months (464.7 ± 316.3 units; *p* = 0.134) but significantly increased after 12 months (484.8 ± 343.6 units; *p* = 0.030) compared to baseline (387.3 ± 264.2 units). On the other hand, anti-CS IgG levels only transiently increased after 6 months (153.5 ± 72.2 units; *p* = 0.041) but then slightly decreased again, still remaining numerically higher after 12 months (145.4 ± 70.1 units; *p* = 0.109) compared to baseline (125.7 ± 61.0 units). Anti-TOPO-F4 IgM levels did not change significantly after 6 months (278.2 ± 147.3 units; *p* = 0.756) and 12 months (307.6 ± 171.1 units; *p* = 0.796) versus baseline (282.6 ± 176.4). Finally, anti-TOPO-F4 IgG autoantibody levels were numerically higher after 6 months (35.6 ± 37.4 units; *p* = 0.092) and 12 months (36.9 ± 46.3 units; *p* = 0.110) in comparison to baseline (24.1 ± 33.0 units). However, the latter differences did not attain significance (Figure 2).

As discussed above, we have previously assessed and published anti-hsp60 data in this cohort. There was no change in anti-hsp60 levels over time in the full RA + AS cohort or in the RA or AS subsets [9].

### 2.4. Correlations of nAAb Levels with Other Parameters

In the simple correlation analysis, in the full RA + AS cohort, baseline anti-CS IgG correlated with baseline BMI (R = 0.412; *p* = 0.013). Baseline anti-TOPO-F4 IgM was inversely correlated with 12-month CRP (R = −0.394; *p* = 0.017). Post-treatment, 12-month anti-TOPO-F4 IgM was associated with RF at baseline (R = 0.602; *p* = 0.005), after 6 months (R = 0.552; *p* = 0.012), and after 12 months (R = 0.543; *p* = 0.013). anti-TOPO-F4 IgG after 6 months exerted a positive correlation with 12-month FMD (R = 0.364; *p* = 0.029) (Appendix A). In the RA subset, baseline (R = 0.501; *p* = 0.025), 6-month (R = 0.490; *p* = 0.028), and 12-month anti-CS IgM (R = 0.456; *p* = 0.043) all positively correlated with baseline FMD. Anti-CS IgM after 12 months was also negatively correlated with 12-month DAS28-CRP (R = −0.499; *p* = 0.025). Finally, anti-TOPO-F4 IgM after 12 months was associated with RF at baseline (R = 0.602; *p* = 0.005), after 6 months (R = 0.552; *p* = 0.012), and after 12 months (R = 0.543; *p* = 0.013) (Appendix A). In the AS subset, anti-CS IgG at baseline showed a positive correlation with BMI (R = 0.676; *p* = 0.004). Anti-CS IgG at baseline (R = −0.495; *p* = 0.047), after 6 months (R = −0.547; *p* = 0.028), and after 12 months (R = −0.597; *p* = 0.015) all showed inverse associations with 12-month PWV. Anti-TOPO-F4 IgM at baseline (R = −0.624; *p* = 0.010), after 6 months (R = −0.501; *p* = 0.048), and after 12 months (R = −0.517; *p* = 0.040) negatively correlated with post-treatment CRP. In addition, 12-month anti-TOPO-F4 IgM was also inversely correlated with 6-month CRP (R = −0.575; *p* = 0.020) and 12-month DAS28-CRP (R = −0.534; *p* = 0.033). Finally, anti-TOPO-F4 IgG after 6 months (R = 0.537; *p* = 0.032) and after 12 months (R = 0.724; *p* = 0.002) showed a positive association with 12-month FMD (Appendix A).

The results of the univariable and multivariable regression analyses are indicated in Table 1. With respect to the univariable analysis, in the full RA + AS cohort, baseline anti-CS IgG determined 6-month CEP autoantibody levels (*p* = 0.013). While anti-TOPO-F4 IgM positively correlated with RF (*p* < 0.05) and inversely with post-treatment FMD (*p* < 0.05) at various time points, baseline anti-TOPO-F4 IgG correlated with baseline (*p* = 0.005), 6-month (*p* = 0.003), and 12-month anti-Hsp60 (*p* < 0.001) (Table 1). Among these correlations, the multivariable analysis confirmed the associations of anti-TOPO-F4 IgG with 6-month (*p* = 0.001) and 12-month anti-Hsp60 (*p* < 0.001) (Table 1). Within the RA subset, baseline anti-CS IgM and IgG positively correlated with 12-month FMD (*p* = 0.010) and 6-month CEP (*p* = 0.013), respectively. In addition, anti-TOPO-F4 IgM variably correlated with RF at different time points (*p* < 0.05), while baseline anti-TOPO-F4 IgG was inversely associated with 6-month PWV (*p* = 0.043) and anti-Hsp60 at all time points (*p* < 0.05) (Table 1). The multivariable analysis did not confirm any significant correlations (Table 1). Finally, in the AS subset, anti-CS IgM at baseline (*p* = 0.005), after 6 months (*p* = 0.040), and 12 months (*p* = 0.039), clearly determined post-treatment disease activity (BASDAI-12). Moreover, anti-TOPO-F4 IgM correlated inversely (*p* < 0.05), while anti-TOPO-F4 IgG positively correlated with 12-month FMD (*p* < 0.05) (Table 1). The negative association between baseline anti-TOPO-F4 IgM (*p* = 0.005) and the positive correlation of 12-month anti-TOPO-F4 IgG (*p* = 0.002) with post-treatment FMD was also confirmed by the multivariable analysis (Table 1).

GLM RM-ANOVA analyzes the effects of nAAbs at baseline together with the 12-month anti-TNF therapy on the 0–6–12-month changes in other parameters. On the other hand, GLM two-way ANOVA shows possible associations between 0–12-month changes in nAAb levels together with TNF inhibition and 0–12-month changes in other parameters. GLM RM-ANOVA and two-way ANOVA results are included in Table 2. In the full cohort, RM-ANOVA revealed that baseline anti-CS IgM and anti-TOPO-F4 IgG levels determine the one-year changes in CRP (*p* = 0.010) and FMD (*p* = 0.032), respectively (Table 2). In the two-way ANOVA, one-year changes in anti-CS IgM, anti-CS IgG, and anti-TOPO-F4 IgM all determined the one-year changes of CRP, disease activity (DAS28-CRP or BASDAI), and PWV (*p* < 0.05). One-year changes in anti-CS IgM were also associated with changes in FMD (*p* < 0.001) (Table 2). In the RA subset, the two-way ANOVA mirrored the association between anti-CS IgM (*p* = 0.002), anti-CS IgG (*p* = 0.044), and anti-TOPO-F4 IgM (*p* = 0.017) changes and one-year CRP changes. In addition, 0–12-month anti-CS IgM changes also correlated with DAS28-CRP changes (*p* = 0.003) (Table 2). Finally, in the AS subset, the RM-ANOVA showed that baseline anti-CS IgM determined one-year BASDAI changes (*p* = 0.047). The two-way ANOVA revealed correlations between 0–12-month changes of anti-CS IgM with CRP (*p* = 0.012), BASDAI (*p* = 0.014), ccIMT (*p* = 0.018), and PWV (*p* = 0.018) changes (Table 2).

When vascular biomarkers were correlated with imaging markers of vascular physiology, BNP-0 showed a positive association with ccIMT-0 (*p* = 0.016). aHsp-60-0 inversely correlated with FMD-12 (*p* = 0.022) and positively with PWV-0 (*p* = 0.040). Both suPAR-0 (*p* = 0.045) and suPAR-12 (*p* = 0.042) were positively associated with PWV-12 (Appendix A).

A GLM RM-ANOVA was performed to assess the combined determinants of vascular biomarker changes over the 12-month period. The change in oxLDL/β2GPI complex levels between baseline and 12 months was determined by the anti-TNF treatment together with higher baseline disease activity (DAS28-CRP/BASDAI-0) (*p* = 0.014). In addition, TNF inhibition and higher ccIMT-0 determined aHsp-60 (*p* = 0.015) and suPAR changes over the one-year period (*p* = 0.041) (Table 2).

Finally, as presented above, there was a slight heterogeneity among patients with respect to taking medications other than anti-TNFs. We found no associations between nAAbs and the intake of MTX, corticosteroids, or NSAIDs.

## 3. Patients and Methods

### 3.1. Patients

Fifty-three patients with inflammatory arthritis (36 RA and 17 AS) selected for the initiation of anti-TNF therapy were enrolled in the study. Patient characteristics are seen in Table 3. The cohort included 34 women and 19 men, with a mean age of 52.0 ± 12.1 (range: 24–83) years. The mean disease duration was 8.5 ± 7.9 (range: 1–44) years, while the mean age at diagnosis was 43.5 ± 12.1 (range: 23–62) years. Exclusion criteria included unstable hypertension (blood pressure > 140/90 mmHg), congestive heart failure, diabetes mellitus, current inflammatory disease other than RA or AS, infectious disease, or renal failure (serum creatinine ≥ 117 mmol/L). None of the patients received aspirin, clopidogrel, heparin, warfarin, or vasoactive drugs at the time of inclusion. Patients with active disease were recruited prior to initiating biological therapy. All patients started on anti-TNF therapy at baseline and received the same biological treatment after one year. Among the 36 RA patients, 20 received etanercept (ETN) at 50 mg/week subcutaneously (SC) and 16 received certolizumab pegol (CZP) (400 mg at 0, 2, and 4 weeks, and thereafter 200 mg every two weeks SC). Altogether, 18 RA patients were treated with ETN and 13 with CZP in combination with methotrexate (MTX) 15–20 mg weekly. The MTX dosing remained stable during the observation period. The other patients received monotherapy. All 17 AS patients received 50 mg/week of ETN monotherapy SC. RA patients did not take DMARDs other than MTX. Altogether, 12 RA and 2 AS patients currently take low-dose (<6 mg/day) methylprednisolone (Table 3). The patients were permitted to use non-steroidal anti-inflammatory drugs (NSAIDs) at a stable dose during the follow-up period. Disease activity was determined by DAS28-CRP and BASDAI in RA and AS, respectively. Joint assessment and the calculation of DAS28 or BASDAI were performed by two investigators (DK and BS).

The study was approved by the Hungarian Scientific Research Council Ethical Committee (approval No. 14804-2/2011/EKU). Written informed consent was obtained from each patient, and assessments were carried out according to the Declaration of Helsinki.

### 3.2. Laboratory Measurements

After overnight fasting, blood samples were taken from the patients for total cholesterol (TC), low-density lipoprotein cholesterol (LDL-C), high-density lipoprotein cholesterol (HDL-C), and triglyceride (TG). Lipids were determined using routine laboratory methods.

Serum high-sensitivity C reactive protein (hsCRP; normal: ≤5 mg/L) and IgM rheumatoid factor (RF; normal: ≤50 IU/mL) were measured by quantitative nephelometry (Cobas Mira Plus-Roche), using CRP and RF reagents (both Dialab). Among ACPA autoantibodies, anti-cyclic citrullinated peptide (anti-CCP) levels were detected in serum samples using a second-generation Immunoscan-RA CCP2 ELISA test (Euro Diagnostica; normal: ≤25 IU/mL).

Anti-citrullinated enolase peptide 1 (CEP-1) IgG was measured in the serum samples using an in-house peptide ELISA in collaboration with the Karolinska Institutet, Stockholm, Sweden. Anti-CEP-1 IgG levels are presented as arbitrary units/mL (AU/mL), based on a standard curve. The cut-off for positivity was >3.7 AU/mL.

All laboratory assessments were performed at baseline as well as 6 and 12 months after treatment initiation.

### 3.3. Determination of nAAbs

Natural Aabs, including anti-CS and anti-TOPO-F4 IgM and IgG levels, were determined by indirect enzyme-linked immunosorbent assay (ELISA) tests performed after manual sample dilution followed by programmed assay execution on the automated Siemens BEP 2000 AdvanceR platform (Siemens AG, Frankfurt, Germany), as described and published previously in more detail [4,5]. Reading was performed at λ = 450/620 nm. Anti-CS and anti-TOPO-F4 IgM and IgG autoantibody levels were expressed in units as published previously [4,5]. We used the recombinant fragment-4 (F4) of topoisomerase I [amino acid (AA) 450–600] or mitochondrial citrate synthase (CS) from porcine heart (Sigma-Aldrich, St. Louis, MO, USA, C3260) as antigen for detection of IgG and IgM nAAbs by in-house indirect ELISA. Briefly, Nunc MaxiSorp™ ELISA plates were coated at a concentration of 2.25 μg/mL of the antigen in coating buffer (Bio-Rad BUF030, Bio-Rad Laboratories Inc., Hercules, CA, USA) (50 μL/well, 4–6 °C, overnight). After washing and blocking, serum samples were incubated at a 100-fold dilution. Standards, blanks, and high and low controls were processed as patient sera. After three washing steps, anti-human IgM or IgG secondary antibody (Dako, Glostrup, Denmark) was incubated for 30 min, followed by TMB substrate for 15 min, and H_2_SO_4_ stop solution (50 μL/well). Automation and reading were performed as described earlier. A five-point dilution series of our in-house anti-CS standard was used for result quantitation, with subsequent four-point sigmoid curve fitting.

Anti-human Hsp60 IgG levels have already been studied in this cohort [9]. In brief, anti-hsp60 IgG levels were measured by an in-house ELISA in collaboration with Semmelweis University, Budapest, as described previously [9]. In brief, plates were coated with 0.1 μg per well of recombinant human Hsp60 (SPP-740; StressGen, Vicoria, BC, Canada). After washing and blocking (phosphate-buffered saline [PBS], 0.5% gelatine), the wells were incubated with 100 μL of serum samples diluted to 1:500 (PBS, 0.5% gelatine, 0.05% Tween 20). Bound anti-Hsp60 antibodies were detected by antihuman IgG peroxidase-labeled antibodies (Sigma, St. Louis, MO, USA) and *o*-phenylene-diamine (Sigma). Optical Density (OD) values were assessed, and concentrations expressed in AU/mL.

### 3.4. Assessment of Vascular Physiology by Ultrasound

Brachial artery FMD was assessed as described before [1,19]. In brief, an ultrasound examination was performed on the right arm using a 10 MHz linear array transducer (ultrasound system: HP Sonos 5500, Hewlett-Packard, Palo Alto, CA, USA) by a single trained sonographer after 30 min of resting in a temperature-controlled room (basal value for FMD). A B-mode longitudinal section was obtained of the brachial artery above the antecubital fossa. To assess FMD, reactive hyperaemia was induced by the release of a pneumatic cuff around the forearm inflated to suprasystolic pressure for 4.5 min. After deflation, the maximal flow velocity and the arterial diameter were continuously recorded at 90 s. Flow velocities, the baseline diameter, and FMD were ECG gated and detected offline. FMD values were expressed as a % change from the baseline (resting) value.

The ccIMT measurements were carried out as described before [1,19]. Briefly, a duplex ultrasound system (HP Sonos 5500, 10 MHz linear array transducer) was used to assess the common carotid arteries by a single observer. Longitudinal high-resolution B-mode ultrasound scans were employed over both right and left common carotid arteries and were R-synchronized and recorded. The offline measurements were performed 1 cm proximal to the carotid bulb in the far wall. ccIMT was defined as the distance between the first and second echogenic lines from the lumen, taking an average of 10 measurements on both sides. Mean ccIMT values were expressed in mm.

With respect to arterial stiffness, PWV was calculated automatically by a TensioClinic arteriograph system (Tensiomed Ltd., Budapest, Hungary) as the quotient of the distance between the jugular fossa and symphysis as described before [1,19,20]. If an artery is elastic, PWV is low. With decreased arterial elasticity, PWV rises. The arteriograph assesses this parameter from the oscillometric data obtained from the 35 mmHg suprasystolic pressure of the brachial artery. To obtain reproducible results, the patient had to rest in a supine position for at least 10 min before the assessment in a quiet room. PWV is expressed in m/s.

### 3.5. Statistical Analysis

Statistical analysis was performed using SPSS version 26.0 (IBM, Armonk, NY, USA) software. Data are expressed as the mean ± SD for continuous variables and percentages for categorical variables. The distribution of continuous variables was evaluated by the Kolmogorov–Smirnov test. Continuous variables were evaluated by a paired two-tailed *t*-test and a Wilcoxon test. Nominal variables were compared between groups using the chi-squared or Fisher’s exact test, as appropriate. Correlations were determined by Pearson’s analysis. Univariable and multivariable regression analyses using the stepwise method were applied to investigate independent associations between dependent and independent variables. The β standardized linear coefficients showing linear correlations between two parameters were determined. The B (+95% CI) regression coefficient indicated independent associations between dependent and independent variables during changes. A general linear model (GLM) repeated measures analysis of variance (RM-ANOVA) was performed to determine the additional effects of nAAbs (an independent variable) on 12-month changes in any other dependent variable. A two-way RM-ANOVA analysis was also conducted to determine the additional effects of nAAb level (an independent variable) changes between baseline and 12 months, together with the effects of treatment, on 12-month changes in any other dependent variable. In the RM-ANOVA and two-way RM-ANOVA analyses, partial η^2^ is given as an indicator of effect size, with values of 0.01 suggesting small, 0.06 medium, and 0.14 large effects [21]. *p* values < 0.05 were considered significant. The reliability of the vascular ultrasound measurements was tested by inter-item correlation and two-way, mixed, single-rater intraclass correlation (ICC [1,3]) before [1,19].

## 4. Discussion

In this study, we evaluated the effects of one-year anti-TNF therapy on the production of nAAbs, including anti-CS and anti-TOPO-F4 IgM and IgG, in the context of vascular pathophysiology. There have been previous publications in the same cohort with respect to CV effects [19], as well as biomarkers including anti-Hsp60 IgG [9]. We do not present previously published materials; however, here we used those data for correlation analysis. Whereas the effects of anti-TNF therapy on vascular pathophysiology have been published in the mixed RA + AS population [19], as new results, we found that one-year TNF-α inhibition transiently improved FMD in RA and decreased PWV in AS. Regarding nAAbs, anti-TNF treatment increased anti-CS and anti-TOPO-F4 levels, and we found several correlations between nAAb levels and other clinical and laboratory parameters.

B-cells produce nAAbs, and nAAbs have been implicated in innate immunity as well as various autoimmune rheumatic diseases and atherosclerosis. Anti-CS and anti-TOPO-F4 IgM and IgM have been detected in autoimmune diseases including SLE, SSc, and RA [4,5], as well as atherosclerosis and ASCVD [7,8]. We and others have also studied the roles of anti-Hsp nAAbs in inflammatory diseases and atherosclerosis [6,9,10,11,12,13,14,15,16,17,18].

In this study, one-year anti-TNF therapy significantly increased anti-CS IgM and IgG, as well as anti-TOPO-F4 IgG, in the mixed RA + AS cohort. With respect to the RA and AS subsets, TNF inhibition increased anti-CS IgM and anti-TOPO-F4 IgG in RA and anti-CS IgM and, transiently, IgG in AS. Moreover, in the various correlation analyses, in summary, anti-CS IgG and anti-TOPO-F4 IgM variably positively correlated with anti-CEP and RF, respectively, indicating the possible role of these nAAbs in autoimmunity. Anti-CS and anti-TOPO-F4 nAAbs are variably and inversely correlated with CRP and indicators of disease activity (DAS28-CRP, BASDAI). With respect to vascular pathophysiology, nAAbs are variably positively correlated with FMD and inversely with PWV at different time points. Specifically, anti-CS IgM correlated with FMD at baseline, while anti-TOPO-F4 IgG correlated after 12 months. This suggests that IgM and IgG nAAb isotypes might influence endothelial function differently during treatment. Moreover, the correlations between nAAbs and FMD were observed in the mixed RA + AS cohort, as well as in RA and AS, while those between nAAbs and PWV were seen only in RA. Finally, we used anti-Hsp60 data previously generated in this very same cohort [9] to correlate this nAAb with other nAAbs, anti-CS and anti-TOPO-F4. Indeed, both in the RA + AS full cohort and the RA subset, anti-Hsp60 IgG at every time point correlated with baseline anti-TOPO-F4 IgG. This suggests that baseline anti-TOPO-F4 IgG might determine anti-Hsp60 over time. Various nAAbs might interact in inflammatory diseases, such as RA. The role of anti-Hsp60 in RA and AS has been established before [9,12,14,15,16,17,18].

RM-ANOVA and two-way ANOVA can analyze correlations between one parameter and changes and between changes of two parameters, respectively. These assessments revealed that baseline anti-CS IgM might be a major determinant of CRP and BASDAI changes during the one-year treatment. This is supported by previous reports on the involvement of nAAbs in various autoimmune rheumatic diseases [4,5]. Moreover, baseline anti-TOPO-F4 IgG highly determined FMD changes over time. This underscores nAAb involvement in vascular pathology and ASCVD [5,7,8]. In addition, 12-month changes in anti-CS and anti-TOPO-F4 IgM and IgG levels variably positively correlated with changes in CRP, RA, and AS disease activity, FMD, PWV, and ccIMT over time. Changes in vascular pathophysiology meant an improvement (increase) of FMD and a decrease of PWV and ccIMT.

Our results depict the context of the biologic treatment of RA and AS, the production of nAAbs, and vascular pathology changes. In this context, anti-TNF therapy decreases systemic inflammation, as indicated by CRP and disease activity. At the same time, TNF inhibition increases nAAb levels. The inverse, independent correlations between nAAbs and CRP or DAS28-CRP/BASDAI over time suggest that nAAbs might exert additive effects on the anti-inflammatory action of biologics. The notion that nAAbs are positively correlated with RF and anti-CEP indicates the continuous involvement of these nAAbs in autoimmunity. With respect to vascular pathology, anti-TNF improved FMD and PWV in this study and preserved ccIMT. The latter means that without treatment, ccIMT would have further deteriorated over time in both RA and AS [1,3,19]. These nAAbs variably and independently correlated with FMD, PWV, and ccIMT improvement, or at least no change. Thus, these nAAbs might also support the beneficial effects of biologic therapy on vascular pathophysiology.

As a further mechanistic explanation of our results, it has been established that during B-cell activation via sIg and CD40, the autocrine production of TNF-α plays an important role in B-cell proliferation [22]. The important characteristic of B1-cells is that they continuously produce nAAbs without antigenic stimulation, which suggests that TNF-α stimulation might not have a significant role in this process. Based on our results, we assume that the nAAb production of B1-cells might be less influenced by TNF-α compared to the antibody production of B2-cells. At the same time, B1-cell subsets showing different TNF-α sensitivity might be responsible for producing nAAbs with different specificities. Due to the anti-inflammatory effects of anti-TNF treatment, the mitochondrial CS released from the cells might decrease; therefore, the continuously produced anti-CS nAAbs in free form might be present in larger quantities. At the same time, it seems that the levels of anti-TOPO-F4 can be less affected by TNF-α inhibition because the free, measurable anti-TOPO-F4 levels did not change significantly in response to treatment. The increase in anti-TOPO-F4 IgG at 12 months is less convincing due to the large SD. However, this change might also be due to anti-TNF treatment. As inflammation decreases upon therapy, fewer inflammatory cells will be destroyed, leading to fewer antigens and more free anti-TOPO-F4 IgG.

Our study might have strengths and limitations. The major strength of the study is that this is the very first report on the effects of any biologic on anti-CS and anti-TOPO-F4 nAAbs. We were also able to put nAAbs and anti-TNF therapy in context with clinical and laboratory parameters of RA and AS, as well as vascular pathophysiology. Thus, we present additional information on the role of nAAbs in autoimmunity, inflammation, and vascular pathology. Limitations might include the relatively small study sample, which might have obscured potentially significant results. In addition, patients with a potentially positive history of ASCVD were also included. Possible limitations also include the heterogeneity in the treatment regimens of the patients, the small number of patients in each group, and the single-center nature of the study.

## 5. Conclusions

Natural autoantibodies, such as anti-CS, anti-TOPO-F4, and anti-Hsp, might play an important role in inflammation, autoimmunity, and atherosclerosis. This is the very first study showing that anti-TNF biologics might increase anti-CS and possibly anti-TOPO-F4 levels in RA and AS. This has been shown in relation to anti-Hsp60 before. TNF inhibition also improves endothelial function and arterial stiffness and might also stabilize carotid atherosclerosis. In this study, anti-TNF therapy increased nAAb levels, including anti-CS IgM and IgG, and possibly also anti-TOPO-F4 IgG. Our correlation analyses suggest that nAAbs might be independently involved in autoimmunity as well as changes in inflammation and vascular pathology over time in biologic-treated patients. We also found associations between anti-TOPO-F4 IgG and anti-Hsp60 IgG. Moreover, nAAb levels or nAAb level changes might be associated with changes in disease activity and markers of vascular pathophysiology over time. Thus, the interplay between arthritis and inflammatory atherosclerosis, as well as the effects of anti-TNF biologics on these pathologies, might independently involve nAAbs. 

## Figures and Tables

**Figure 1 ijms-25-03429-f001:**
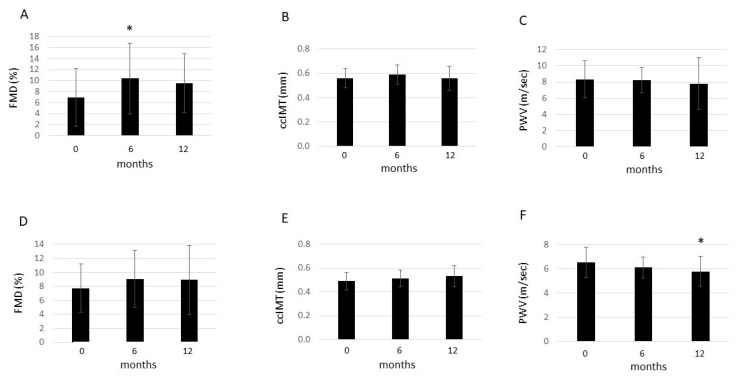
Effects of 1-year anti-TNF therapy on FMD, ccIMT, and PWV in RA (**A**–**C**) and AS (**D**–**F**). Asterisks indicate significant changes (*p* < 0.05).

**Figure 2 ijms-25-03429-f002:**
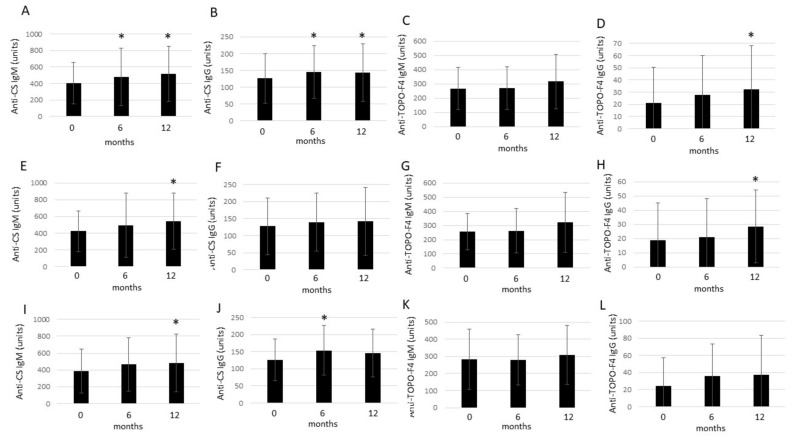
Effects of 1-year anti-TNF therapy on anti-CS IgM, anti-CS IgG, anti-TOPO-F4 IgM, and anti-TOPO-F4 IgG in the RA + AS full cohort (**A**–**D**), as well as in the RA (**E**–**H**) and AS subset (**I**–**L**). Asterisks indicate significant changes (*p* < 0.05).

**Table 1 ijms-25-03429-t001:** Univariable and multivariable regression analysis of the effects of natural autoantibodies as independent variables on other dependent variables.

Full (RA + AS) Cohort
Dependent Variable	Independent Variable	Univariable Analysis	Multivariable Analysis
β	CI 95%	B	*p*	β	CI 95%	B	*p*
RF-6	TOPO-F4 IgM-0	0.477	0.215–4.658	2.437	0.033	
RF-12	TOPO-F4 IgM-12	0.511	0.499–5.458	2.979	0.021
CEP-6	CS IgG-0	0.586	0.037–0.275	0.156	0.013
FMD-12	TOPO-F4 IgM-0	0.394	0.013–0.026	0.070	0.017
TOPO-F4 IgM-6	0.343	0.003–0.016	−0.054	0.040
HSP60-0	TOPO-F4 IgG-0	0.458	0.139–0.719	0.429	0.005
HSP60-6	TOPO-F4 IgG-0	0.485	0.167–0.731	0.449	0.003	0.492	0.188–0.785	0.456	0.001
HSP60-12	TOPO-F4 IgG-0	0.543	0.240–0.799	0.519	<0.001	0.549	0.265–0.811	0.522	<0.001
RA subset
Dependent variable	Independent variable	Univariable analysis	Multivariable analysis
β	CI 95%	B	*p*	β	CI 95%	B	*p*
RF-0	TOPO-F4 IgM-0	0.464	0.134–4.713	2.423	0.039	
RF-6	TOPO-F4 IgM-0	0.477	0.215–4.658	2.437	0.033
RF-12	TOPO-F4 IgM-12	0.511	0.499–5.458	2.979	0.021
CEP-6	CS IgG-0	0.586	0.037–0.275	0.156	0.013
FMD-12	CS IgM-0	0.455	0–0.019	0.010	0.044
PWV-6	TOPO-F4 IgG-0	−0.456	−0.011–0	−0.006	0.043
TOPO-F4 IgG-6	−0.465	−0.009–0	−0.005	0.039
HSP60-0	TOPO-F4 IgG-0	0.469	0.029–0.831	0.430	0.037
HSP60-6	TOPO-F4 IgG-0	0.624	0.276–1.173	0.724	0.003
HSP60-12	TOPO-F4 IgG-0	0.646	0.332–1.266	0.799	0.002
AS subset
Dependent variable	Independent variable	Univariable analysis	Multivariable analysis
β	CI 95%	B	*p*	β	CI 95%	B	*p*
BASDAI-12	CS IgM-0	0.666	0.001–0.004	0.002	0.005	
CS IgM-6	0.517	0–0.003	0.002	0.040
CS IgM-12	0.520	0–0.003	0.001	0.039
FMD-12	TOPO-F4 IgM-0	0.578	0.017–0.157	0.087	0.019	0.515	0.027–0.127	0.077	0.005
TOPO-F4 IgM-6	0.514	0.003–0.133	0.068	0.041	
TOPO-F4 IgG-6	0.548	0.002–0.035	0.018	0.028
TOPO-F4 IgG-12	0.655	0.006–0.032	0.019	0.006	0.601	0.008–0.027	0.017	0.002

**Table 2 ijms-25-03429-t002:** General linear model (GLM) repeated measurements ANOVA (RM-ANOVA) and two-way ANOVA analyses of the effects of natural autoantibodies and natural autoantibody changes on 0–12 month changes in other parameters.

Full (RA + AS) Cohort
RM-ANOVA	Two-Way ANOVA
Dependent Variable	Effect	F	Partial η^2^	*p*	Dependent Variable	Effect	F	Partial η^2^	*p*
CRP (0–12)	CS IgM-0	4.885	0.126	0.010	CRP (0–12)	CS IgM (0–12)	20.980	0.375	<0.001
FMD (0–12)	TOPO-F4 IgG-0	3.617	0.096	0.032	CS IgG (0–12)	10.186	0.225	0.003
	TOPO-F4 IgM (0–12)	16.208	0.317	<0.001
DAS28-CRP/BASDAI (0–12)	CS IgM (0–12)	2.013	0.364	<0.001
CS IgG (0–12)	6.458	0.156	0.016
TOPO-F4 IgM (0–12)	7.227	0.171	0.011
FMD (0–12)	CS IgM (0–12)	18.315	0.344	<0.001
PWV (0–12)	CS IgM (0–12)	19.252	0.355	<0.001
CS IgG (0–12)	5.119	0.128	0.030
TOPO-F4 IgM (0–12)	5.120	0.128	0.030
RA subset
RM-ANOVA	Two-way ANOVA
Dependent variable	Effect	F	partial η^2^	*p*	Dependent variable	Effect	F	partial η^2^	*p*
	CRP (0–12)	CS IgM (0–12)	12.277	0.393	0.002
CS IgG (0–12)	4.673	0.197	0.044
TOPO-F4 IgM (0–12)	6.903	0.267	0.017
DAS28-CRP (0–12)	CS IgM (0–12)	11.941	0.986	0.003
AS subset
RM-ANOVA	Two-way ANOVA
Dependent variable	Effect	F	partial η^2^	*p*	Dependent variable	Effect	F	partial η^2^	*p*
BASDAI (0–12)	CS IgM-0	3.911	0.376	0.047	CRP (0–12)	CS IgM (0–12)	8.263	0.355	0.012
	BASDAI (0–12)	CS IgM (0–12)	7.640	0.337	0.014
ccIMT (0–12)	CS IgM (0–12)	15.000	0.318	0.018
PWV (0–12)	CS IgM (0–12)	7.074	0.320	0.018

**Table 3 ijms-25-03429-t003:** Patient characteristics.

	RA	AS	Total
*n*	36	17	53
female:male	31:5	3:14	34:19
age (mean ± SD) (range), years	55.9 ± 9.8 (35–83)	43.6 ± 12.4 (24–72)	52.0 ± 12.1 (24–83)
disease duration (mean ± SEM) (range), years	9.1 ± 8.3 (1–44)	7.2 ± 7.0 (1–26)	8.5 ± 7.9 (1–44)
age at diagnosis	47.0 ± 8.7 (28–62)	36.4 ± 11.6 (23–50)	43.5 ± 12.1 (23–62)
smoking (current)	7	7	14
positive CV history	8	1	9
BMI (mean ± SD), kg/m^2^	29.3 ± 3.6	31.1 ± 3.8	29.9 ± 3.7
obesity (BMI > 30 kg/m^2^)	17	11	28
diabetes mellitus history	3	1	4
hypertension history	17	4	21
RF positivity, *n* (%)	26 (72)	-	-
Anti-CCP positivity, *n* (%)	21 (58)	-	-
DAS28-CRP (baseline) (mean ± SD)	5.00 ± 0.86	-	-
BASDAI (baseline) (mean ± SD)	-	5.79 ± 1.19	-
Treatment (ETN, CZP)	20 ETN, 16 CZP	17 ETN	37 ETN, 16 CZP

## Data Availability

Data are available from authors upon request.

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
