# Peer review of "Natural Autoantibodies in Biologic-Treated Rheumatoid Arthritis and Ankylosing Spondylitis Patients: Associations with Vascular Pathophysiology"

_ijms, 2024, doi:10.3390/ijms25063429_

Round 1

Reviewer 1 Report

Comments and Suggestions for Authors

The manuscript entitled " Effects of one-year anti-TNF-α therapy on natural autoantibodies in association with vascular pathophysiology in rheumatoid arthritis and ankylosing spondylitis" has very important topic widely discussed  in the last 5 years. 

I have several recommendations to the authors:

1. I would recommend to the authors  to check the following information in the section patients and methods -  " Among the 36 RA patients, 20 received etanercept (ETN) 50 mg/week subcutaneous (SC) and 16 received certolizumab pegol (CZP) (400 mg at 0, 2 and 4 weeks, and thereafter 200 mg twice weekly SC)"  Are the patients on CZP are really on dosage 200mg twice weekly? In my opinion they are on regular dose 200mg in two weeks period or 400mg every four weeks.

2. Clarify the dose of MTX  - "Altogether 18 RA patients were treated with ETN and 13 with CZP in combination with methotrexate (MTX)"

3. Include additional information about the NSAIDS. Whether or not this patients take NSAIDS in stable dose?

4. Include information about the joint assessment of the patients ( and who performed it), because DAS-28 needs this information.

5. " The offline measurements were performed 1 cm proximal to the carotid bulb in the far wall. ccIMT was defined as the distance between the first and second echogenic lines from the lumen taking the average of 10 measurements on both sides. ccIMT values were expressed in mm." How many measurements the ultrasonographer performed (3 and take the median measurement or one)? 

6. The figures and tables are with good quality.

7. Include in the whole text that it is DAS28-CRP. Especially in results part.

8. Authors can include in the limitations of the study the heterogenеity in the treatment regimens of the patients and small number of patients in each group. Also, another limitation of the study if it is a single center collecting the patients. 

The manuscript is well written with detailed information about each stage of the study being conducted. Congratulations to the authors for the great work.

Comments on the Quality of English Language

Minor edition

Author Response

Dear Sirs

we thank Reviewer 1 for thoroughly assessing our manuscript, and also for the helpful comments. We tried to address all issues raised and revised the manuscript accordingly. We made the requested changes in RED colour.

  1. I would recommend to the authors  to check the following information in the section patients and methods -  " Among the 36 RA patients, 20 received etanercept (ETN) 50 mg/week subcutaneous (SC) and 16 received certolizumab pegol (CZP) (400 mg at 0, 2 and 4 weeks, and thereafter 200 mg twice weekly SC)"  Are the patients on CZP are really on dosage 200mg twice weekly? In my opinion they are on regular dose 200mg in two weeks period or 400mg every four weeks.

No, we are sorry but ours is the recommended dosing. We use 400 mgs three times as loading dose, and the maintenance dose is 200 mg every two weeks. So our mistake is that we wrote twice weekly instead of every two weeks. Otherwise tha 400 mg every 4 weeks is an alternative regimen, which we do not use in our institution. We corrected the dosing.

  1. Clarify the dose of MTX  - "Altogether 18 RA patients were treated with ETN and 13 with CZP in combination with methotrexate (MTX)"

The dose was between 15-20 mg/week, and we did not change the dose during the observation period. This info is now included.

  1. Include additional information about the NSAIDS. Whether or not this patients take NSAIDS in stable dose?

Various NSAID products were used in a stable dose. This info is now included.

  1. Include information about the joint assessment of the patients (and who performed it), because DAS-28 needs this information.

Joint assessment and DAS28 calculation were performed by two investigators (DK and BS). This information is now included.  

  1. " The offline measurements were performed 1 cm proximal to the carotid bulb in the far wall. ccIMT was defined as the distance between the first and second echogenic lines from the lumen taking the average of 10 measurements on both sides. ccIMT values were expressed in mm." How many measurements the ultrasonographer performed (3 and take the median measurement or one)? 

As written, an average of 10 measurements were performed on both sides. The presented values are means.

  1. The figures and tables are with good quality.

Many thanks.

  1. Include in the whole text that it is DAS28-CRP. Especially in results part.

Many thanks, now it is included. We also added DAS28-CRP to tables 1 and 3.

  1. Authors can include in the limitations of the study the heterogenеity in the treatment regimens of the patients and small number of patients in each group. Also, another limitation of the study if it is a single center collecting the patients. 

We added these limitations.

The manuscript is well written with detailed information about each stage of the study being conducted. Congratulations to the authors for the great work.

Many thanks for the positive response and opinion.

Reviewer 2 Report

Comments and Suggestions for Authors

The article presents the possible effect of anti-TNF alpha therapy on natural antibody levels concerning cardiovascular events. It is well-written and requires minor revision. I also recommend taking into consideration the following remarks:

-        I recommend changing the title of the article to enhance the article's visibility and relevance as it does not present the main theme of the article

-        rephrase “We and others have also linked anti-Hsp nAAbs to inflammation, autoimmunity and atherosclerosis [6, 9-18].”

-        rephrase “Although the increase in anti-TOPO-F4 IgG at 12 months is less convincing due to the large SD, yet it might still be explained because of anti-TNF treatment”

-        provide, if possible, the stages of the diseases included at the moment of biological therapy beginning

-        integrate information regarding the other drugs taken by the patients and if there are any correlations between them and the studied antibodies. If not consider it as a limitation of the study

-        consider developing the conclusion section

Author Response

Dear Sirs

we thank Reviewer 2 for thoroughly assessing our manuscript, and also for the helpful comments. We tried to address all issues raised and revised the manuscript accordingly. We made the requested changes in RED colour.

  1. I recommend changing the title of the article to enhance the article's visibility and relevance as it does not present the main theme of the article

We changed the title as requested.

  1. rephrase “We and others have also linked anti-Hsp nAAbs to inflammation, autoimmunity and atherosclerosis [6, 9-18].”

Rephrased.

  1. rephrase “Although the increase in anti-TOPO-F4 IgG at 12 months is less convincing due to the large SD, yet it might still be explained because of anti-TNF treatment”

Rephrased.

  1. provide, if possible, the stages of the diseases included at the moment of biological therapy beginning

We provided data on mean disease duration and range (8.5±7.9 (range: 1-44) years). Unfortunately we cannot present more detailed data.

  1. integrate information regarding the other drugs taken by the patients and if there are any correlations between them and the studied antibodies. If not consider it as a limitation of the study

As also requested by Reviewer 1, we now added more information on MTX and NSAIDs and added the heterogeneity of drug treatment as limitations. The studied antibodies showed no associations with MTX, corticosteroid and NSAID intake. This information is now included in the Results.

  1. consider developing the conclusion section

We further developed the Conclusions.

Round 2

Reviewer 1 Report

Comments and Suggestions for Authors

I don't have any further recommendaations.